# Evaluating the Energy and Core Nutrients of Condiments in China

**DOI:** 10.3390/nu15204346

**Published:** 2023-10-12

**Authors:** Wanting Lv, Xin Ding, Yang Liu, Aiguo Ma, Yuexin Yang, Zhu Wang, Chao Gao

**Affiliations:** 1Institute of Nutrition and Health, School of Public Health, Qingdao University, Qingdao 266071, China; lvwanting0827@163.com (W.L.); dingxin19960222@163.com (X.D.); magfood@126.com (A.M.); yuexin_yang@sina.com (Y.Y.); 2Key Laboratory of Trace Element Nutrition of National Health Commission, National Institute for Nutrition and Health, Chinese Center for Disease Control and Prevention, 29 Nanwei Road, Beijing 100050, China; liuyang@ninh.chinacdc.cn (Y.L.); wzhblue@163.com (Z.W.); 3Chinese Nutrition Society, Beijing 100053, China

**Keywords:** condiments, nutrition labelling, sodium, sodium reduction, food component

## Abstract

Condiments are a significant source of sodium in the diets of Chinese residents. This study aimed to analyze the nutrient content of China’s major condiments and to provide support for setting the reference intake for condiments in order to take measures on salt reduction in China. Nutrition data for condiments were collected from the Database on Nutrition Labelling of Prepackaged Foods China in 2017–2022, and by online access to food composition databases from France, the UK, Belgium, and Japan. The analyses include 1510 condiments in China and 1565 related condiments in four countries, of which the descriptive indicators were examined such as median, IQR, and range. Cross-comparisons were made in terms of the difference between the content levels in five countries and the “WHO global sodium benchmarks”. The results show that among the 15 types of condiments in China, sesame/peanut butter-based products have a relatively high content of energy, fat, and protein, namely, 2580 kJ/100 g, 50 g/100 g, and 22.2 g/100 g, respectively. In addition to salt, chicken extract/chicken powder, bouillon cubes, and soy sauce are also high in sodium. Furthermore, there were significant differences in the contents of energy and core nutrients across different products when benchmarking with similar condiments in the five countries (*p* < 0.001). The sodium content and fat content of some condiments are excessively high. Therefore, enhancing residents’ consumption awareness and reducing the amount of condiments is of great significance for reducing China’s per capita salt intake and promoting good health.

## 1. Introduction

Excessive dietary intakes of energy, sodium, fat, and sugar increase the risk of obesity, hypertension, diabetes, and other related non-communicable chronic diseases [1,2]. A Systematic Analysis of the Global Burden of Disease Study suggested that in 2017, approximately 11 million deaths and 255 million disability-adjusted life years globally were attributable to dietary risk factors [3], with high sodium intake leading to approximately 3 million deaths [4].Controlling the intake of nutrients that implicate public health, such as sodium, is one of the most cost-effective and feasible ways to improve health and reduce the burden of non-communicable diseases [5].

To press for the solution of unhealthy diets and strengthen the responsibility of the food and non-alcoholic beverage industry in addressing the burden of non-communicable diseases, in 2022, the World Health Organization (WHO) published the “WHO global sodium benchmarks” for different food categories and called on food operators to implement them globally, in an effort to complement national and regional measures of setting sodium targets and facilitate countries to develop national policies [6]. The Chinese government proposed the “Healthy China” strategy, and has issued a series of nutrition and health policies, such as the “National Nutrition plan (2017–2030)” in 2017 and the “Healthy China action (2019–2030)” in 2019 [7]. Relevant studies have shown that the most important measure to improve the health of Chinese residents is to improve their awareness of dietary knowledge [8], so as to reduce dietary risks and control the intake of oil, salt, sugar/sweeteners, and other ingredients in the diet.

Currently, the majority of dietary sodium (75–80%) in Western diets comes from processed, packaged, and prepared foods [9]. In contrast, according to a study on the dietary sodium intake of adult residents in 15 provinces in China in 2018, only 10.1% of the sodium in the dietary diet of Chinese residents comes from processed foods, while the rest mainly comes from condiments added during the cooking process. The proportion of condiments with an intake exceeding 2000 mg/d reaches 86.7% [10]. Internationally, condiments are defined as substances used to enhance the flavor of food, such as mustard, tomato ketchup, salt, and spices [11]; China’s national standard (GB/T 20903-2007) defines condiments as products that are widely used in catering, cooking, and food processing to harmonize taste and odor and have the effect of removing fishy, stinky, and greasy aspects, while adding flavor and freshness, and include salt, sugar, soy sauce, monosodium glutamate, sauces, compound seasonings, etc. [12]. Relevant research studies have shown that the demand for condiments has continued to grow over the past decade, making it the second-largest specialty food segment after cheese [13]. An international study of the top 10 countries using sauces, spices, and condiments shows that the main condiments consumed in China are soy sauces and bouillon cubes; the main condiments consumed in the USA are spices; the main condiments consumed in Vietnam, Thailand, and Myanmar are fish sauce [14], and condiments play an important role in both pre-packaged processed foods and catering dishes [15].

Although a few previous studies have aimed to determine the health status of prepackaged foods and the distribution of sodium content in sauces in China, no specific analysis has been made on the current status of the nutrient levels of relevant condiments [16,17,18]. Therefore, this study conducted a descriptive statistical analysis of the nutrient content of existing condiments on the market, evaluated the current distribution of their content levels, and selected similar products from the national food databases of France, UK, Belgium, and Japan to compare and observe the differences in the contents of various nutrients. This study also examined the difference between the sodium content in Chinese condiments and the global sodium content benchmark, to inform the thresholds applicable to the nutrient level of condiments in China. Specific direction for salt reduction policies in the future was proposed, to provide reference data and theoretical basis in guiding the healthy and reasonable consumption of Chinese consumers as well as in the reformulation of condiments by food manufacturers and production enterprises.

## 2. Materials and Methods

### 2.1. Data Collection of Chinese Condiments

#### 2.1.1. Sample Collection

The data for the study are all from the nutrition label database of prepackaged foods in China, which was established in accordance with the relevant provisions of China’s “General Principles for Nutrition Labeling of Prepackaged Foods (GB 28050-2011) [19]”. It contains information on nutrition labeling of major prepackaged foods collected by the Institute for Nutrition and Health of the Chinese Center for Disease Control and Prevention from 19 provinces, including Beijing, Xinjiang Production and Construction Corps, Jiangxi, Jilin, Hebei, and Hainan, in 2017–2022. Nutrition label photos were taken to collect, record, and sort out the main content data, and duplicates were deleted through duplicate checking; by referring to the net content of the product; the portion value was standardized and converted into mg/100 g and mL/100 g.

#### 2.1.2. Sample Inclusion Criteria

In order to ensure the representativeness and credibility of the market survey data on prepackaged food, the following criteria were established for data collection [20]:

Those with higher consumption among residents; those with clearly-indicated product information such as product name, factory address, and formula; those that conform to basic logic, that is, the sum of the energy converted from fat, carbohydrate, and protein on the label does not exceed the energy value on the label.

#### 2.1.3. Sample Classification

According to the “Food Composition Data Expression Specification” published by the former National Health and Family Planning Commission of China, “Condiment Classification (GB/T 20903-2007)”, “National Food Safety Standard Food Additive Use Standard (GB 2760-2014) [21]” and Catalogue for Food Production Licensing, after taking into account the characteristics of various condiments, eating habits, production processes, formulas, and nutrient distribution, etc., the research team classified the condiments step by step [22]. To this end, the data on condiments in China was divided into 15 categories, of which paste and like products, compound seasoning, and hot pot seasoning had further sub-categories (Appendix A). The specific number and distribution of various condiments is shown in Figure 1.

### 2.2. Data Collection of France, UK, Belgium, and Japan Condiments

The data of condiments comes from the following sources: France [23]—Oqali database, which provides most of the information available on processed food packaging on the French market (Nutritional composition, content, …); UK [24]—McCance and Widdowson’s Comprehensive Food Ingredient Dataset (CoFID) (2021), which covers 2886 foods and ingredients, and has helped food manufacturers, dietitians, software and app producers access nutritional information for food and recipes; Belgium [25]—VoG Nubel database, which provides information of 6000 branded products (in terms of energy content, fat content, sodium content and other nutrient content); Japan [26]—STANDARD TABLES DF FOOD COMPOSITION IN JAPAN-2015-(Seventh Revised Version) (Appendix A).

The following fields of information were extracted: Universal Product Code (UPC), brand name, labelling unit (g/100 g), product description, full ingredient list, energy (kJ/100 g), protein (g/100 g), fat (g/100 g), carbohydrate (g/100 g), and sodium (mg/100 g). 

In light of the original food classification of various countries, the researchers cross-referenced the condiment categories of China and decided on the relevant data to be included in the study (Appendix A). The details are shown in Figure 2.

### 2.3. Statistical Analysis

The Kolmogorov–Smirnov test (K-S test) was used to test the normality of the data. As the data did not follow the normal distribution, the median, IQR and range were used to analyze the distribution of energy (kJ/100 g), protein (g/100 g), fat (g/100 g), carbohydrate (g/100 g), and sodium (mg/100 g) for each sub-category of condiments in China in a descriptive way.

The Kruskal–Wallis H test was used to compare the differences in content composition between similar condiments in France, UK, Belgium, and Japan. If the difference was statistically significant, post hoc tests were carried out using the Bonferroni correction. A *p* value of <0.05 was considered statistically significant. The analyses were conducted using IBM SPSS V.26.0.

## 3. Results

### 3.1. Nutrient Distribution in Different Categories of Condiments in China

According to our analysis, sesame paste and peanut butter have a relatively high energy and fat content compared to other products, with 2580 kJ/100 g and 50 g/100 g, respectively; second, the hot pot base follows with an energy content of 2400 kJ/100 g and a fat content of 55 g/100 g. Mayonnaise/salad dressing is not to be overlooked as a high energy and fat condiment with a median energy content of 1750 kJ/100 g and the fat content second only to sesame/peanut butter with a median content of 38.3 g/100 g. In terms of protein, sesame paste/peanut butter has the highest protein content of all the condiment, with a median content of 22.2 g/100 g. Following closely are chicken essence/chicken powder products, with 22.0 g/100 g due to their raw material factors. Among all condiments, salt, chicken essence, and bouillon cube ingredients have the top three sodium contents, which are 38,737 mg/100 g, 19,000 mg/100 g, and 7392.52 mg/100 g respectively.

### 3.2. Comparison with Selected Condiments in France, UK, Belgium, and Japan

This study compared certain nutrients between Chinese and French, British, Belgian, and Japanese condiments. The comparison results are shown in Figure 3. The results are as follows:

For spicy agent products, the energy and sodium content of Chinese products are lower than that of Japan (energy: 1423.5 kJ/100 g vs. 1558.5 kJ/100 g, *p* = 0.005; sodium: 25.5 mg/100 g vs. 56.0 mg/100 g, *p* = 0.038); For bouillon cube products, the sodium content of Chinese products is higher than that of the UK (7392.5 mg/100 g vs. 3740 mg/100 g, *p* = 0.005); For soy sauce products, there are significant differences between China and Japan and Belgium in the content of energy, protein, fat, carbohydrate, and sodium (*p* < 0.001). The sodium content of soy sauce in China is higher than Japan and Belgium (6600 mg/100 g vs. 3300 mg/100 g vs. 5476 mg/100 g, *p* < 0.001). For tomato ketchup products, the carbohydrate content of Chinese products is higher than that of France (26.25 g/100 g vs. 6.7 g/100 g, *p* < 0.001), and the sodium content lower than France (939 mg/100 g vs. 2000 mg/100 g, *p* = 0.006), but higher than Belgium (939 mg/100 g vs. 655 mg/100 g, *p* < 0.001). For mayonnaise/salad dressing products, a cross-comparison between China and France, the UK and Belgium showed that the energy, fat, and sodium content of the UK products are lower, with a median content of 689 kJ/100 g, 11.2 g/100 g, and 212.5 mg/100 g respectively (*p* < 0.001), while French products have the highest fat content (72.85 g/100 g) (*p* < 0.001). For compound seasoning products, the comparison of condiments in China, UK, France, Belgium, and Japan showed that Chinese products have higher energy and protein contents (*p* < 0.001), and French products have a higher fat content (25 g/100 g) (*p* < 0.001).

### 3.3. Comparison with “WHO Global Sodium Benchmarks” Levels

WHO published “WHO global sodium benchmarks” in September 2021, in comparison among Chinese condiments, the median sodium contents of mayonnaise/salad dressing (762 mg/100 g), tomato ketchup (939 mg/100 g), pickled vegetable (1970 mg/100 g), and soy sauce (6600 mg/100 g) are much higher than the WHO global sodium benchmarks, while the sodium content of bouillon cubes (7392.5 mg/100 g) is lower than WHO global sodium benchmarks (15,000 mg/100 g). In terms of the distribution of sodium content in these five product categories, the proportion of Chinese mayonnaise/salad dressings that exceeded the sodium benchmark value was 90.7%, followed by soy sauce at 89.3% and bouillon cubes at a relatively low 32.0%. The comparison results are shown in Figure 4.

## 4. Discussion

Compared with the published articles, this study provides a more detailed and comprehensive descriptive statistical analysis of Chinese condiments. The results show, sesame paste/peanut butter is high in energy, fat, and protein, while salt, chicken essence, and bouillon cubes are the top three in terms of actual sodium content in Chinese condiments. Second, it can be seen in comparison with the distribution of content levels of similar products in France, the UK, Belgium, and Japan, that the protein content and fat content of Chinese mayonnaise/salad dressing do not differ significantly from those of Japan, but the sodium content is significantly higher than that of the UK and France. There are significant differences in the distribution of nutrient content between the different countries in the category of compound seasonings, which may be due to the fact that seasonings are commonly consumed worldwide and countries have different taste preferences, consumer behavior habits, etc. Finally, in comparison with the “WHO global sodium benchmarks”, it can be seen that the “WHO global sodium benchmarks” are not entirely suitable as a reference standard for sodium reduction in Chinese condiments, while the sodium content of Chinese soy sauce exceeds the sodium benchmark by 89.3%, suggesting still more room for sodium reduction.

According to an analysis of the sodium content in 4082 commodities in China [27], the food group with the highest average sodium content (6888.6 mg/100 g) was “sauces, dressings, springs, and dips”. Not only China, but also many countries in the world currently have a high sodium content. In 2022, a cross-sectional study on sodium content and labelling of packaged foods was carried out in Nigeria [28], which showed the median sodium content of “sauces, dressings, spreads, and dips” was 536 mg/100 g; a survey and study conducted in Fiji showed that the median sodium content of “sauces, dressings, spreads, and dips” was 670 mg/100 g [29]; The median sodium content in the category “sauces/dressings” in packaged foods in Colombia was 1533.3 mg/100 g [30]. These contents all exceeded the “WHO global sodium benchmarks” of 500 mg/100 g, which shows that the high sodium content of condiments is a problem deserving to be taken seriously. Moreover, in addition to salt, mayonnaise/salad dressing, peanut butter, and compound seasoning products, there are some other condiments with unique traditional characteristics in China, and their sodium contents are also notably high; for example, the median sodium content of fermented bean curd is 3500 mg/100 g, the median sodium content of hotpot seasoning is 4000 mg/100 g, both of which are popular condiments in daily life and are in the upper middle level of all condiments in terms of sodium content, thus they also need to be taken into account.

As a globally consumed food product, condiments are an important food flavor enhancer [31]. In particular, salt and soy sauce are the main sources of dietary sodium intake among Chinese residents [32]. As dietary patterns change, there is an increasing concern at the global level on the effect of dietary sodium on health, as excessive dietary sodium intake increases blood pressure and increases the risk of hypertension, cardiovascular disease, stroke, and chronic kidney disease [33,34]. The WHO at the World Health Assembly actively called for “a relative reduction of salt intake by 30% by 2025” [35], and the Healthy China Action Plan for 2019–2030 also recommended consuming salt less than 5 g/d [36]. However, China currently has one of the highest salt intakes in the world, with an average adult salt intake of 11 g/day [37], and daily cooking salt consumption per capita is 9.3 g/day, well above the WHO recommended intake. According to the results of this study, in the daily diet, in addition to focusing on sodium in salt, attention should also be paid to sodium in high sodium condiments, such as chicken essence/chicken powder, bouillon cubes, soy sauce, etc. According to Table 1, the distribution of the sodium content of soy sauce, bouillon cubes, and oyster sauce/fish sauce is 3140–10,120 mg/100 g, 3910–27,460 mg/100 g, 2386.7–10,000 mg/100 g, and the distribution of fat content of fermented bean curd, mayonnaise/salad dressing, and hot pot seasoning is 0.5–52 g/100 g, 11.1–76.7 g/100 g, 5.4–87.8 g/100 g. The wide range of sodium and fat content distribution in these products suggests that there is much room for sodium and fat reduction. At the same time, it also implies that when formulating the corresponding standards, such as the guidelines for salt reduction, we can consider making the standards for these categories appropriately stringent to better practice China’s relevant nutrition policies. It is worth mentioning that the consumption pattern of condiments is different from ordinary food, and different condiments complement each other, which will lead to a multiplication of salt intake in total. Therefore, it is of great importance to map the actual content of condiments and to establish effective evaluation standards suitable for condiments in China in light of their flavor, texture, safety, stability, and consumer acceptance, in a timely manner, and adopt strategic measures to reduce salt gradually.

Reviewing the salt reduction strategies of relevant countries, the UK has reduced the salt content of many food categories by 20–50% in 10 years since the introduction of its salt reduction scheme in 2003, and condiment is one of them. The scheme promotes progressive reformulation of targeted foods on a voluntary basis to reduce per capita salt intake [38]. In the process, five sets of salt reduction targets were established in 2006, 2009, 2011, 2014, and 2020, proving effective a target-oriented approach in reducing the sodium content of many foods [39]. In France, since the introduction of the “Nutri-Score” five-color FOPL system in 2017, it has improved the situation of different socio-economic groups and incentivized manufacturers to reformulate their existing and new products, thus successfully improving the nutrition profiles of condiments [40,41]. In order to reduce the total dietary sodium intake of the Chinese population, promote a healthy diet, and gear the food industry towards nutrition, a series of salt reduction initiatives have also been carried out in China. In 2015, the Institute for Nutrition and Health of the Chinese Center for Disease Control and Prevention, together with the Chinese Nutrition Society, set up a working group on salt reduction guidelines. Built on the guidelines on salt reduction from the food industry of many countries, with repeated discussions, opinions consultation, and comment collection from enterprises, experts in various fields, managers, and policy makers, the working group eventually proposed in 2019 the basic objectives and related contents of salt reduction in China’s food industry by different food categories, with the release of the China Food Industry Salt Reduction Guidelines. This is of great significance in reducing the total dietary sodium intake of Chinese residents and guiding the new direction of dietary culture [42].

All the data of this study come from the nutrition label database of prepackaged food of the Institute of Nutrition and Health of the Chinese Center for Disease Control and Prevention, which covers the main condiments currently circulating in the Chinese market. This study for the first time carefully divides the condiments into 15 categories and adopts standardized methods for related data collection and processing, e.g., standardized food coding, food classification, etc., to ensure the authenticity and comparability of the data, showing practical significance. 

It is worth noting that in comparing similar condiments between China and France, the UK, Belgium, and Japan, public databases and relevant articles from each country were thoroughly searched with our best efforts. However, the specific relevant product information is not as detailed as in China. Therefore, data collection for the remaining countries is limited, and may not include all condiment labelling values from the selected countries, which may be considered a limitation. In the same sense, the results of the study and analysis may not be nationally representative of all condiments in the above countries, but are enough to provide a reference for best estimates of commonly used condiments in each country. Finally, our research group will also continue to collect and analyze data on the content of condiments in China in the future, aiming to effectively update and monitor the concrete implementation of salt reduction initiatives in China, provide data and information to support the development of salt reduction targets for relevant categories of products in China, and improve consumer purchasing preferences and the development of industrial technology for the products. In conclusion, raising awareness of condiments and reducing the use of related condiments in food preparation and industrial production is of great importance in reducing the per capita salt intake and promoting the health of our population.

## 5. Conclusions

Among the condiments currently available in China, sesame paste/peanut butter products, which are high in energy and fat, and chicken essence/chicken powder and soy sauce, which are high in sodium, need to be given priority for attention. In addition, high sodium levels are a key health concern. The five countries listed in the paper all have different salt reduction policies for the foods on the market, where the results of the study have shown that developing strategies based on salt reduction targets can be effective in reducing sodium levels in food. Therefore, there is an urgent need to set clear targets for feasible or further sodium reduction, actively work towards a rigorous reformulation program, as well as conduct regular monitoring on relevant condiments to understand how they may be changed with government actions, changes in people’s preferences, or other long-term trends.

## Figures and Tables

**Figure 1 nutrients-15-04346-f001:**
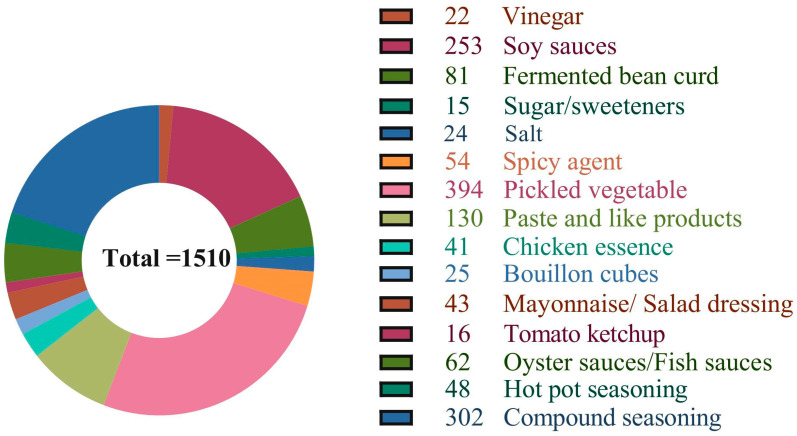
Distribution of Chinese Condiments.

**Figure 2 nutrients-15-04346-f002:**
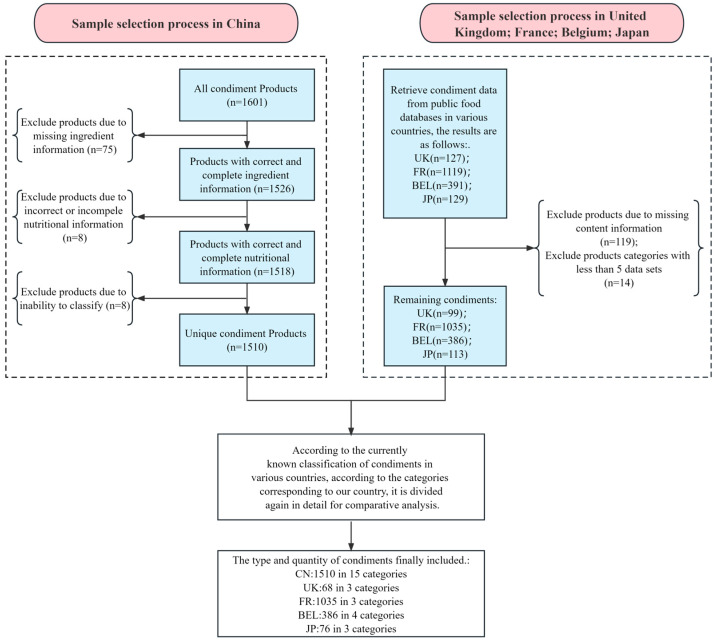
Sample selection process. CN: China; UK: The United Kingdom; FR: France; BEL: Belgium; JP: Japan.

**Figure 3 nutrients-15-04346-f003:**
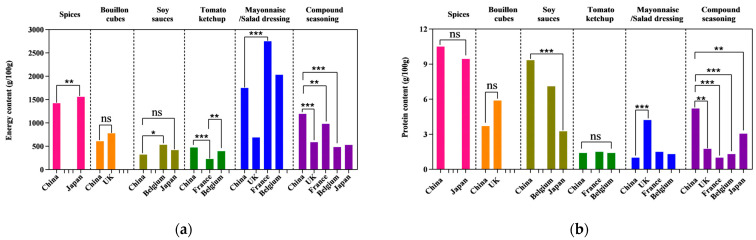
Chinese condiments vs. the some condiments in France, UK, Belgium, and Japan. (**a**) Comparison of energy content; (**b**) comparison of protein content; (**c**) comparison of fat content; (**d**) comparison of carbohydrate content; (**e**) comparison of sodium content. The “*” means *p* < 0.05. The “**” means *p* < 0.01. The “***” means *p* < 0.001. The “ns” means no significance.

**Figure 4 nutrients-15-04346-f004:**
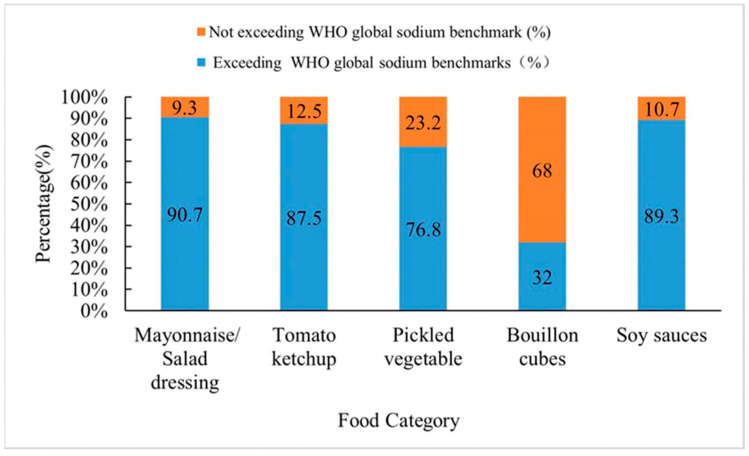
Sodium content of condiments in China vs. “WHO global sodium benchmarks”. Reference “WHO global sodium benchmarks”. Mayonnaise/salad dressing—500 mg/100 g; tomato ketchup—650 mg/100 g; marinades and thick pastes—1425 mg/100 g; bouillon cubes and soup stock powders—15,000 mg/100 g; soy sauce—4840 mg/100 g.

**Table 1 nutrients-15-04346-t001:** Nutrient distribution of condiments in China.

	Food Category		n	Energy (kJ/100 g)			Protein (g/100 g)			Fat (g/100 g)			Carbohydrate (g/100 g)			Sodium (mg/100 g)		
				Median	IQR	Range	Median	IQR	Range	Median	IQR	Range	Median	IQR	Range	Median	IQR	Range
1	Vinegar		22	119	75.75–159.25	0 *–330	3.05	1.55–7.2	0–8.1	0	0–0	0–0.1	1.65	0–4.1	0–18	320	101.25–557.5	0–1100
2	Soy sauces		253	320	224–410	40–959	9.33	6.67–11	0.4–15.33	0	0–0	0 *–1.2	8	5.33–13	0–220	6600	5861.5–7260	3140–10,120
3	Fermented bean curd		81	720	536.5–859.5	300–2124	10	9.15–13	7–19.7	10	6.7–15	0.5–52	3.9	1.95–9.4	0–26.3	3500	3000–3851	2080–5859
4	Sugar/sweeteners		15	1490	1326–1628	1264–1693	0.4	0–1.15	0–3.2	0	0–0.05	0–5	93	68.5–97.65	62–99.6	10	0–32.5	0–106
5	Salt		24	0	0–0	0–600	0	0–0	0–6.7	0	0–0	0–8.9	0	0–0	0–66.5	38,737	30,167.75–38,909	16,300–39,311
6	Spicy agent (cayenne pepper, sprinkling pepper)		54	1424	804.75–1542.25	116–219	10.5	6.43–15	0–35.2	4.85	2.5–7.53	0–41.1	60.95	15.58–70.2	2.2–81.1	25.5	10–100.25	0–3636
7	Pickled vegetable		394	230	123–434.25	46–2492	1.7	1.3–2.4	0–28	2.15	0.6–6.5	0–60	4.5	2.2–7.5	0–52.4	1970	1490–2360	12–12,646
8	Paste and like products		130	595	457.5–2206.08	87–2987	6.68	4.8–16	2.2–93.33	5.2	3.3–30.33	0–59	14.6	10.5–19.78	2.4–166.67	4462	1186–7182.5	0–12,000
		Chinese bean paste	101	543	440–690.5	87–2987	5.8	4.35–8.1	2.2–93.33	4.2	2.6–6.55	0–53	13.7	9.35–17.55	2.4–166.67	5413	3669–7881.5	550–12,000
		Sesame paste/Peanut butter	29	2580	2541–2600	2233.33–2866	22.2	19.2–24.15	6–26.8	50	46.95–52.7	31.33–59	16.8	15.35–21.1	10–57.33	39	10–356.5	0–520
9	Chicken essence		41	960	870–1020	497–1224	22	10–24	1.6–47	4	1.75–6	0–12.8	27.8	22–32	1.5–300	19,000	16,486–19,880	12,400–22,000
10	Bouillon cubes		25	607	452.99–1010.5	172–1889.47	3.7	2.70–6.55	0–11.3	5.6	2.05–11.91	0–39.9	10.8	5.92–23.8	2.7–70.4	7392.52	4775–17,084.5	3910–27,460
11	Mayonnaise/Salad dressing		43	1750	1308–2550	582–3001	1	0.67–2	0–9.9	38.3	28.1–63.4	11.1–76.7	11.3	6.2–18.7	1.6–31	762	598–1200	318–2321
12	Tomato ketchup		16	473	464–496.5	81–568.42	1.4	1.09–1.5	0–4.9	0	0–0.15	0–0.7	26.25	24.55–26.85	14–29.47	939	777.5–1031	37–1378.95
13	Oyster sauces/Fish sauces		62	330	240–498.33	75–969.23	3.3	2.67–4.17	0.6–12.6	0	0–0	0–2.8	13	10.53–24.17	0–53.85	4366.7	3987.75–5066.67	2366.7–10,000
14	Hot pot seasoning		48	1591	984–2709.25	577–3276	5.1	3–7.35	1–14.9	29.2	15.63–68	5.4–87.8	11	3.43–18.23	0.6–49.5	4000	2960–5660	1707–9110
		Hot pot base	35	2400	1348–2830	899–3276	4.2	2.2–6	1–14.9	55	25.8–71.2	5.4–87.8	8	2.6–16.8	0.6–49.5	4740	3970–6390	2250–9110
		Hot pot Dipping Sauce	13	768	730.5–1017.5	577–1346	7.1	6.4–9.45	4.7–13.4	9.1	8.5–16.2	6.3–26.9	12.8	9.45–20.3	4–21.1	2727	2468–2897.5	1707–3111
15	Compound seasoning		302	1194	428–2012	48–3244	5.2	2.5–8.55	0–56	13.14	1.73–41	0–80	12.8	6.38–25.13	0–71.43	2671.5	1460–4607.5	28–21,000
		Other solid compound seasoning	61	1469	873–1723	120–2264	12.8	6.8–17.65	0–56	10.7	3.2–19.45	0–39.1	42.2	22.55–50.7	1.1–69.9	4630	2813.5–8773.5	28–21,000
		Sauce based on vegetable ingredients	147	1150	351–2396.5	48–3244	4.4	1.98–7.1	0–22	22.45	1.5–55.63	0–80	8.45	5.58–14.9	0–71.43	1900	1277–3414	120–8150
		Sauce based on animal ingredients	30	1664	1000–2291	187–3164	7.2	4.1–10.95	1.1–25.4	37.55	15.4–48.3	3.5–76	10.4	5.12–15.8	2.3–43	1800	1176.5–2474	300–5008.89
		Other semi-solid compound sauces	35	780	520–1443	208–2961	3.8	1.4–6.4	0 *–17.1	9.3	0.9–27.3	0 *–72.1	14.5	9.1–32.2	2.8–63.57	4586	2949–5088	339–8528.57
		Seasoned green seasoning	29	413	262–656.5	60–1801	3.4	2.6–4.45	0–16.67	0	0–2.67	0 *–40.4	14.7	10–21.1	1.7–50	4066	2565–5990.5	536–10,000

Notes: * General Principles for Nutrition Labeling of Prepackaged Foods (GB 28050-2011) in China stipulates that when the value of a nutrient content is ≤0 threshold, its content should be labelled as “0”. “IQR” means interquartile range.

## Data Availability

Data sets generated during the study are available from the corresponding author on reasonable request.

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
