# Peer review of "Evaluating the Energy and Core Nutrients of Condiments in China"

_nutrients, 2023, doi:10.3390/nu15204346_

Round 1

Reviewer 1 Report

Comments to the Authors

Strategies to reduce chronic non-communicable diseases through the reduction of salt content in the diet and other components is a worldwide company. The standards established by the World Health Organization make it possible to delineate patterns and steps to follow in different countries to reach optimal healthy consumption. However, despite these efforts, many countries are today, well above this recommended value. In this context, this type of study provides scientific tools to be able to address this problem and prop up the critical scales in this matter, in different countries. Taking into account, that the sodium intake in China mainly comes from the condiments added during the cooking process, the work carries out a systematic and strict analysis of the content of the different condiments used in China (classified into 15 categories) and compares them with others present in other countries. This study shows that many of the products of daily consumption in China present levels higher than those established by the WHO in energy and salt content. Likewise, it is important to highlight that many of the foods do not have a complete description of the content, which is why they have not been included in the study.

This manuscript has been well described, using a friendly organization and description, however, the authors should pay attention to some aspects:

Table 1, place horizontally or in some way that the content reads better.

Figure 3, enlarges the size of the figure because the axes and the statistics are not read well. Add a symbol to notice the statistical differences between countries.  It is difficult to interpret without reading the result section.

In the legend of Figure 3 and Table 1, place a detail of the statistic used, indicating its significance.

The legend of figure 4 should include a further description for its interpretation without reading the text.

It could be discussed, if the authors know, if there is a relationship between products with excess energy and salt and their commercial impact. If the brand of some sells more than the others (wide range of nutrients in the same category of products).

Some information included in the S1 supplementary material could be included in the manuscript and discussed along with current consumption from these countries. As possible strategies to follow for countries that begin with this process, since, as the authors mentioned, the cultural factor is key in dietary regulation.

Discuss the wide range of each nutrient in each category of condiment products in China, in relation to the need to adjust the established guidelines to reduce the consumption of salt, fat, etc. in that country.

Reviewer 2 Report

This is a good-quality manuscript. The research problem is relevant and important. The applied method is correct. The results are presented in the appropriate way and the implications are properly discussed.

Please add separate sections on Limitations and Future Research Directions.

Please refer to:

Bryła P., Selected predictors of the importance attached to salt content information on the food packaging (a study among Polish consumers), Nutrients, 2020, Vol. 12, 293. https://doi.org/10.3390/nu12020293.

Wang, Z., Zhu, Z., Cai, H., Luo, B., Shi, Z., Liu, Y., ... & Su, J. (2022). The high sodium condiments and pre-packaged food should be the focus of dietary sodium control in the adult Shanghai population. Nutrition & Metabolism19(1), 1-8.

Gao, T., Huang, X., Chen, X., Cai, X., Huang, J., Vincent, G., & Wang, S. (2023). Advances in flavor peptides with sodium-reducing ability: A review. Critical Reviews in Food Science and Nutrition, 1-17.

Chailek, C., Thitichai, P., Praekunatham, H., Taweewigyakarn, P., & Chantian, T. (2023). Peer Reviewed: Availability and Price of Low-Sodium Condiments and Instant Noodles in the Bangkok Metropolitan Region. Preventing Chronic Disease20.

lines 36 and 43 - non-communicable

line 64 - please explain what d in mg/d means

71 - research studies

158 - please explain IQR

233 - more detailed than what?

303 - remove "in France"

acceptable
